# Consistent Kernel Mean Estimation
# for Functions of Random Variables

**Carl-Johann Simon-Gabriel**[*], **Adam Ścibior**[*,†], **Ilya Tolstikhin, Bernhard Schölkopf**
Department of Empirical Inference, Max Planck Institute for Intelligent Systems
Spemanstraße 38, 72076 Tübingen, Germany
[*] joint first authors; [†] also with: Engineering Department, Cambridge University
cjsimon@, adam.scibior@, ilya@, bs@tuebingen.mpg.de

## Abstract

We provide a theoretical foundation for non-parametric estimation of functions of random variables using kernel mean embeddings. We show that for any continuous function $f$, consistent estimators of the mean embedding of a random variable $X$ lead to consistent estimators of the mean embedding of $f(X)$. For Matérn kernels and sufficiently smooth functions we also provide rates of convergence.

Our results extend to functions of multiple random variables. If the variables are dependent, we require an estimator of the mean embedding of their joint distribution as a starting point; if they are independent, it is sufficient to have separate estimators of the mean embeddings of their marginal distributions. In either case, our results cover both mean embeddings based on i.i.d. samples as well as "reduced set" expansions in terms of dependent expansion points. The latter serves as a justification for using such expansions to limit memory resources when applying the approach as a basis for probabilistic programming.

## 1  Introduction

A common task in probabilistic modelling is to compute the distribution of $f(X)$, given a measurable function $f$ and a random variable $X$. In fact, the earliest instances of this problem date back at least to Poisson (1837). Sometimes this can be done analytically. For example, if $f$ is linear and $X$ is Gaussian, that is $f(x) = ax + b$ and $X \sim \mathcal{N}(\mu; \sigma)$, we have $f(X) \sim \mathcal{N}(a\mu + b; a\sigma)$. There exist various methods for obtaining such analytical expressions (Mathai, 1973), but outside a small subset of distributions and functions the formulae are either not available or too complicated to be practical.

An alternative to the analytical approach is numerical approximation, ideally implemented as a flexible software library. The need for such tools is recognised in the general programming languages community (McKinley, 2016), but no standards were established so far. The main challenge is in finding a good approximate representation for random variables.

Distributions on integers, for example, are usually represented as lists of $(x_i, p(x_i))$ pairs. For real valued distributions, integral transforms (Springer, 1979), mixtures of Gaussians (Milios, 2009), Laguerre polynomials (Williamson, 1989), and Chebyshev polynomials (Korzeń and Jaroszewicz, 2014) were proposed as convenient representations for numerical computation. For strings, probabilistic finite automata are often used. All those approaches have their merits, but they only work with a specific input type.

There is an alternative, based on Monte Carlo sampling (Kalos and Whitlock, 2008), which is to represent $X$ by a (possibly weighted) sample $\{(x_i, w_i)\}_{i=1}^n$ (with $w_i \geq 0$). This representation has several advantages: (i) it works for any input type, (ii) the sample size controls the time-accuracy trade-off, and (iii) applying functions to random variables reduces to applying the functions pointwise

to the sample, i.e., $\{(f(x_i), w_i)\}$ represents $f(X)$. Furthermore, expectations of functions of random variables can be estimated as $\mathbb{E}[f(X)] \approx \sum_i w_i f(x_i) / \sum_i w_i$, sometimes with guarantees for the convergence rate.

The flexibility of this Monte Carlo approach comes at a cost: without further assumptions on the underlying input space $\mathcal{X}$, it is hard to quantify the accuracy of this representation. For instance, given two samples of the same size, $\{(x_i, w_i)\}_{i=1}^n$ and $\{(x_i', w_i')\}_{i=1}^n$, how can we tell which one is a better representation of $X$? More generally, how could we optimize a representation with predefined sample size?

There exists an alternative to the Monte Carlo approach, called Kernel Mean Embeddings (KME) (Berlinet and Thomas-Agnan, 2004; Smola et al., 2007). It also represents random variables as samples, but additionally defines a notion of similarity between sample points. As a result, (i) it keeps all the advantages of the Monte Carlo scheme, (ii) it includes the Monte Carlo method as a special case, (iii) it overcomes its pitfalls described above, and (iv) it can be tailored to focus on different properties of $X$, depending on the user's needs and prior assumptions. The KME approach identifies both sample points and distributions with functions in an abstract Hilbert space. Internally the latter are still represented as weighted samples, but the weights can be negative and the straightforward Monte Carlo interpretation is no longer valid. Schölkopf et al. (2015) propose using KMEs as approximate representation of random variables for the purpose of computing their functions. However, they only provide theoretical justification for it in rather idealised settings, which do not meet practical implementation requirements.

In this paper, we build on this work and provide general theoretical guarantees for the proposed estimators. Specifically, we prove statements of the form "if $\{(x_i, w_i)\}_{i=1}^n$ provides a good estimate for the KME of $X$, then $\{(f(x_i), w_i)\}_{i=1}^n$ provides a good estimate for the KME of $f(X)$". Importantly, our results *do not assume joint independence* of the observations $x_i$ (and weights $w_i$). This makes them a powerful tool. For instance, imagine we are given data $\{(x_i, w_i)\}_{i=1}^n$ from a random variable $X$ that we need to compress. Then our theorems guarantee that, whatever compression algorithm we use, as long as the compressed representation $\{(x_j', w_j')\}_{j=1}^n$ still provides a good estimate for the KME of $X$, the pointwise images $\{(f(x_j'), w_j')\}_{j=1}^n$ provide good estimates of the KME of $f(X)$.

In the remainder of this section we first introduce KMEs and discuss their merits. Then we explain why and how we extend the results of Schölkopf et al. (2015). Section 2 contains our main results. In Section 2.1 we show consistency of the relevant estimator in a general setting, and in Section 2.2 we provide finite sample guarantees when Matérn kernels are used. In Section 3 we show how our results apply to functions of multiple variables, both interdependent and independent. Section 4 concludes with a discussion.

## 1.1 Background on kernel mean embeddings

Let $\mathcal{X}$ be a measurable input space. We use a positive definite bounded and measurable kernel $k : \mathcal{X} \times \mathcal{X} \to \mathbb{R}$ to represent random variables $X \sim P$ and weighted samples $\hat{X} := \{(x_i, w_i)\}_{i=1}^n$ as two functions $\mu_X^k$ and $\hat{\mu}_X^k$ in the corresponding Reproducing Kernel Hilbert Space (RKHS) $\mathcal{H}_k$ by defining

$$\mu_X^k := \int k(x, .) \, dP(x) \quad \text{and} \quad \hat{\mu}_X^k := \sum_i w_i k(x_i, .) \,.$$

These are guaranteed to exist, since we assume the kernel is bounded (Smola et al., 2007). When clear from the context, we omit the kernel $k$ in the superscript. $\mu_X$ is called the KME of $P$, but we also refer to it as the KME of $X$. In this paper we focus on computing functions of random variables. For $f : \mathcal{X} \to \mathcal{Z}$, where $\mathcal{Z}$ is a measurable space, and for a positive definite bounded $k_z : \mathcal{Z} \times \mathcal{Z} \to \mathbb{R}$ we also write

$$\mu_{f(X)}^{k_z} := \int k_z(f(x), .) \, dP(x) \quad \text{and} \quad \hat{\mu}_{f(X)}^{k_z} := \sum_i w_i k_z(f(x_i), .) \,. \tag{1}$$

The advantage of mapping random variables $X$ and samples $\hat{X}$ to functions in the RKHS is that we may now say that $\hat{X}$ is a good approximation for $X$ if the RKHS distance $\|\hat{\mu}_X - \mu_X\|$ is small. This distance depends on the choice of the kernel and different kernels emphasise different information about $X$. For example if on $\mathcal{X} := [a, b] \subset \mathbb{R}$ we choose $k(x, x') := x \cdot x' + 1$, then

$\mu_X(x) = \mathbb{E}_{X \sim P}[X]\, x + 1$. Thus any two distributions and/or samples with equal means are mapped to the same function in $\mathcal{H}_k$ so the distance between them is zero. Therefore using this particular $k$, we keep track only of the mean of the distributions. If instead we prefer to keep track of all first $p$ moments, we may use the kernel $k(x, x') := (x \cdot x' + 1)^p$. And if we do not want to loose any information at all, we should choose $k$ such that $\mu^k$ is injective over all probability measures on $\mathcal{X}$. Such kernels are called *characteristic*. For standard spaces, such as $\mathcal{X} = \mathbb{R}^d$, many widely used kernels were proven characteristic, such as Gaussian, Laplacian, and Matérn kernels (Sriperumbudur et al., 2010, 2011).

The Gaussian kernel $k(x, x') := e^{-\frac{\|x - x'\|^2}{2\sigma^2}}$ may serve as another good illustration of the flexibility of this representation. Whatever positive bandwidth $\sigma^2 > 0$, we do not lose any information about distributions, because $k$ is characteristic. Nevertheless, if $\sigma^2$ grows, all distributions start looking the same, because their embeddings converge to a constant function 1. If, on the other hand, $\sigma^2$ becomes small, distributions look increasingly different and $\hat{\mu}_X$ becomes a function with bumps of height $w_i$ at every $x_i$. In the limit when $\sigma^2$ goes to zero, each point is only similar to itself, so $\hat{\mu}_X$ reduces to the Monte Carlo method. Choosing $\sigma^2$ can be interpreted as controlling the degree of smoothing in the approximation.

## 1.2 Reduced set methods

An attractive feature when using KME estimators is the ability to reduce the number of expansion points (i.e., the size of the weighted sample) in a principled way. Specifically, if $\hat{X}' := \{(x'_j, 1/N)\}_{j=1}^N$ then the objective is to construct $\hat{X} := \{(x_i, w_i)\}_{i=1}^n$ that minimises $\|\hat{\mu}_{X'} - \hat{\mu}_X\|$ with $n < N$. Often the resulting $x_i$ are mutually dependent and the $w_i$ certainly depend on them. The algorithms for constructing such expansions are known as *reduced set methods* and have been studied by the machine learning community (Schölkopf and Smola, 2002, Chapter 18).

Although reduced set methods provide significant efficiency gains, their application raises certain concerns when it comes to computing functions of random variables. Let $P, Q$ be distributions of $X$ and $f(X)$ respectively. If $x'_j \sim_{i.i.d.} P$, then $f(x'_j) \sim_{i.i.d.} Q$ and so $\hat{\mu}_{f(X')} = \frac{1}{N} \sum_j k(f(x'_j), .)$ reduces to the commonly used $\sqrt{N}$-consistent empirical estimator of $\mu_{f(X)}$ (Smola et al., 2007). Unfortunately, this is not the case after applying reduced set methods, and it is not known under which conditions $\hat{\mu}_{f(X)}$ is a consistent estimator for $\mu_{f(X)}$.

Schölkopf et al. (2015) advocate the use of reduced expansion set methods to save computational resources. They also provide some reasoning why this should be the right thing to do for characteristic kernels, but as they state themselves, their rigorous analysis does not cover practical reduced set methods. Motivated by this and other concerns listed in Section 1.4, we provide a generalised analysis of the estimator $\hat{\mu}_{f(X)}$, where we do not make assumptions on how $x_i$ and $w_i$ were generated.

Before doing that, however, we first illustrate how the need for reduced set methods naturally emerges on a concrete problem.

## 1.3 Illustration with functions of two random variables

Suppose that we want to estimate $\mu_{f(X,Y)}$ given i.i.d. samples $\hat{X}' = \{x'_i, 1/N\}_{i=1}^N$ and $\hat{Y}' = \{y'_j, 1/N\}_{j=1}^N$ from two independent random variables $X \in \mathcal{X}$ and $Y \in \mathcal{Y}$ respectively. Let $Q$ be the distribution of $Z = f(X, Y)$.

The first option is to consider what we will call the *diagonal estimator* $\hat{\mu}_1 := \frac{1}{N} \sum_{i=1}^n k_z(f(x'_i, y'_i), .)$. Since $f(x'_i, y'_i) \sim_{i.i.d.} Q$, $\hat{\mu}_1$ is $\sqrt{N}$-consistent (Smola et al., 2007). Another option is to consider the *U-statistic* estimator $\hat{\mu}_2 := \frac{1}{N^2} \sum_{i,j=1}^N k_z(f(x'_i, y'_j), .)$, which is also known to be $\sqrt{N}$-consistent. Experiments show that $\hat{\mu}_2$ is more accurate and has lower variance than $\hat{\mu}_1$ (see Figure 1). However, the U-statistic estimator $\hat{\mu}_2$ needs $O(n^2)$ memory rather than $O(n)$. For this reason Schölkopf et al. (2015) propose to use a reduced set method both on $\hat{X}'$ and $\hat{Y}'$ to get new samples $\hat{X} = \{x_i, w_i\}_{i=1}^n$ and $\hat{Y} = \{y_j, u_j\}_{j=1}^n$ of size $n \ll N$, and then estimate $\mu_{f(X,Y)}$ using $\hat{\mu}_3 := \sum_{i,j=1}^n w_i u_j k_x(f(x_i, y_j), .)$.

We ran experiments on synthetic data to show how accurately $\hat{\mu}_1, \hat{\mu}_2$ and $\hat{\mu}_3$ approximate $\mu_{f(X,Y)}$ with growing sample size $N$. We considered three basic arithmetic operations: multiplication $X \cdot Y$, division $X/Y$, and exponentiation $X^Y$, with $X \sim \mathcal{N}(3; 0.5)$ and $Y \sim \mathcal{N}(4; 0.5)$. As the true embedding $\mu_{f(X,Y)}$ is unknown, we approximated it by a U-statistic estimator based on a large sample (125 points). For $\hat{\mu}_3$, we used the simplest possible reduced set method: we randomly sampled subsets of size $n = 0.01 \cdot N$ of the $x_i$, and optimized the weights $w_i$ and $u_i$ to best approximate $\hat{\mu}_X$ and $\hat{\mu}_Y$. The results are summarised in Figure 1 and corroborate our expectations: (i) all estimators converge, (ii) $\hat{\mu}_2$ converges fastest and has the lowest variance, and (iii) $\hat{\mu}_3$ is worse than $\hat{\mu}_2$, but much better than the diagonal estimator $\hat{\mu}_1$. Note, moreover, that unlike the U-statistic estimator $\hat{\mu}_2$, the reduced set based estimator $\hat{\mu}_3$ can be used with a fixed storage budget even if we perform a sequence of function applications—a situation naturally appearing in the context of probabilistic programming.

Schölkopf et al. (2015) prove the consistency of $\hat{\mu}_3$ only for a rather limited case, when the points of the reduced expansions $\{x_i\}_{i=1}^n$ and $\{y_i\}_{i=1}^n$ are i.i.d. copies of $X$ and $Y$, respectively, and the weights $\{(w_i, u_i)\}_{i=1}^n$ are constants. Using our new results we will prove in Section 3.1 the consistency of $\hat{\mu}_3$ under fairly general conditions, even in the case when both expansion points and weights are interdependent random variables.

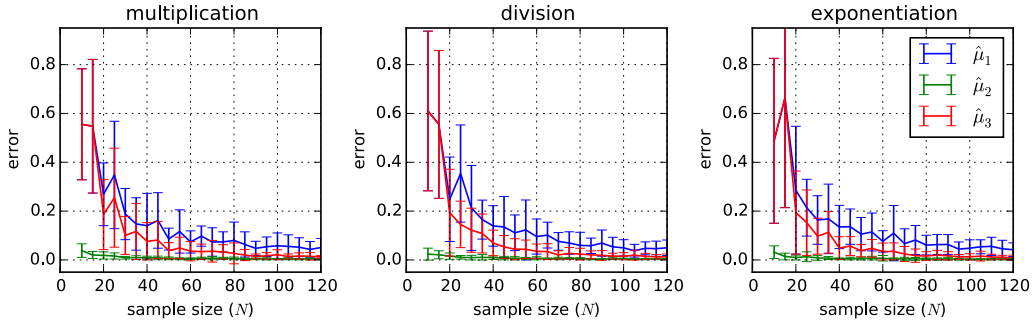

Figure 1: Error of kernel mean estimators for basic arithmetic functions of two variables, $X \cdot Y$, $X/Y$ and $X^Y$, as a function of sample size $N$. The $U$-statistic estimator $\hat{\mu}_2$ works best, closely followed by the proposed estimator $\hat{\mu}_3$, which outperforms the diagonal estimator $\hat{\mu}_1$.

## 1.4 Other sources of non-i.i.d. samples

Although our discussion above focuses on reduced expansion set methods, there are other popular algorithms that produce KME expansions where the samples are not i.i.d. Here we briefly discuss several examples, emphasising that our selection is not comprehensive. They provide additional motivation for stating convergence guarantees in the most general setting possible.

An important notion in probability theory is that of a conditional distribution, which can also be represented using KME (Song et al., 2009). With this representation the standard laws of probability, such as sum, product, and Bayes' rules, can be stated using KME (Fukumizu et al., 2013). Applying those rules results in KME estimators with strong dependencies between samples and their weights.

Another possibility is that even though i.i.d. samples are available, they may not produce the best estimator. Various approaches, such as kernel herding (Chen et al., 2010; Lacoste-Julien et al., 2015), attempt to produce a better KME estimator by actively generating pseudo-samples that are not i.i.d. from the underlying distribution.

## 2 Main results

This section contains our main results regarding consistency and finite sample guarantees for the estimator $\hat{\mu}_{f(X)}$ defined in (1). They are based on the convergence of $\hat{\mu}_X$ and avoid simplifying assumptions about its structure.

## 2.1 Consistency

If $k_x$ is $c_0$-*universal* (see Sriperumbudur et al. (2011)), consistency of $\hat{\mu}_{f(X)}$ can be shown in a rather general setting.

**Theorem 1.** *Let $\mathcal{X}$ and $\mathcal{Z}$ be compact Hausdorff spaces equipped with their Borel $\sigma$-algebras, $f : \mathcal{X} \to \mathcal{Z}$ a continuous function, $k_x, k_z$ continuous kernels on $\mathcal{X}, \mathcal{Z}$ respectively. Assume $k_x$ is $c_0$-universal and that there exists $C$ such that $\sum_i |w_i| \leq C$ independently of $n$. The following holds:*

$$\text{If} \quad \hat{\mu}_X^{k_x} \to \mu_X^{k_x} \quad \text{then} \quad \hat{\mu}_{f(X)}^{k_z} \to \mu_{f(X)}^{k_z} \quad \text{as} \quad n \to \infty.$$

*Proof.* Let $P$ be the distribution of $X$ and $\hat{P}_n = \sum_{i=1}^n w_i \delta_{x_i}$. Define a new kernel on $\mathcal{X}$ by $\widetilde{k}_x(x_1, x_2) := k_z\big(f(x_1), f(x_2)\big)$. $\mathcal{X}$ is compact and $\{\hat{P}_n \,|\, n \in \mathbb{N}\} \cup \{P\}$ is a bounded set (in total variation norm) of finite measures, because $\|\hat{P}_n\|_{TV} = \sum_{i=1}^n |w_i| \leq C$. Furthermore, $k_x$ is continuous and $c_0$-universal. Using Corollary 52 of Simon-Gabriel and Schölkopf (2016) we conclude that: $\hat{\mu}_X^{k_x} \to \mu_X^{k_x}$ implies that $\hat{P}$ converges weakly to $P$. Now, $k_z$ and $f$ being continuous, so is $\widetilde{k}_x$. Thus, if $\hat{P}$ converges weakly to $P$, then $\hat{\mu}_X^{\widetilde{k}_x} \to \mu_X^{\widetilde{k}_x}$ (Simon-Gabriel and Schölkopf, 2016, Theorem 44, Points (1) and (iii)). Overall, $\hat{\mu}_X^{k_x} \to \mu_X^{k_x}$ implies $\hat{\mu}_X^{\widetilde{k}_x} \to \mu_X^{\widetilde{k}_x}$. We conclude the proof by showing that convergence in $\mathcal{H}_{\widetilde{k}_x}$ leads to convergence in $\mathcal{H}_{k_z}$:

$$\left\| \hat{\mu}_{f(X)}^{k_z} - \mu_{f(X)}^{k_z} \right\|_{k_z}^2 = \left\| \hat{\mu}_X^{\widetilde{k}_x} - \mu_X^{\widetilde{k}_x} \right\|_{\widetilde{k}_x}^2 \to 0.$$

For a detailed version of the above, see Appendix A. $\qquad\square$

The continuity assumption is rather unrestrictive. All kernels and functions defined on a discrete space are continuous with respect to the discrete topology, so the theorem applies in this case. For $\mathcal{X} = \mathbb{R}^d$, many kernels used in practice are continuous, including Gaussian, Laplacian, Matérn and other radial kernels. The slightly limiting factor of this theorem is that $k_x$ must be $c_0$-universal, which often can be tricky to verify. However, most standard kernels—including all radial, non-constant kernels—are $c_0$-universal (see Sriperumbudur et al., 2011). The assumption that the input domain is compact is satisfied in most applications, since any measurements coming from physical sensors are contained in a bounded range. Finally, the assumption that $\sum_i |w_i| \leq C$ can be enforced, for instance, by applying a suitable regularization in reduced set methods.

## 2.2 Finite sample guarantees

Theorem 1 guarantees that the estimator $\hat{\mu}_{f(X)}$ converges to $\mu_{f(X)}$ when $\hat{\mu}_X$ converges to $\mu_X$. However, it says nothing about the speed of convergence. In this section we provide a convergence rate when working with Matérn kernels, which are of the form

$$k_x^s(x, x') = \frac{2^{1-s}}{\Gamma(s)} \|x - x'\|_2^{s-d/2} \, \mathcal{B}_{d/2-s} \left( \|x - x'\|_2 \right) \,, \tag{2}$$

where $\mathcal{B}_\alpha$ is a modified Bessel function of the third kind (also known as Macdonald function) of order $\alpha$, $\Gamma$ is the Gamma function and $s > \frac{d}{2}$ is a smoothness parameter. The RKHS induced by $k_x^s$ is the Sobolev space $\mathcal{W}_2^s(\mathbb{R}^d)$ (Wendland, 2004, Theorem 6.13 & Chap.10) containing $s$-times differentiable functions. The finite-sample bound of Theorem 2 is based on the analysis of Kanagawa et al. (2016), which requires the following assumptions:

**Assumptions 1.** *Let $X$ be a random variable over $\mathcal{X} = \mathbb{R}^d$ with distribution $P$ and let $\hat{X} = \{(x_i, w_i)\}_{i=1}^n$ be random variables over $\mathcal{X}^n \times \mathbb{R}^n$ with joint distribution $S$. There exists a probability distribution $Q$ with full support on $\mathbb{R}^d$ and a bounded density, satisfying the following properties:*

 *(i) $P$ has a bounded density function w.r.t. $Q$;*
 *(ii) there is a constant $D > 0$ independent of $n$, such that*

$$\underset{S}{\mathbb{E}} \left[ \frac{1}{n} \sum_{i=1}^n g^2(x_i) \right] \leq D \|g\|_{\mathrm{L}^2(Q)}^2 \,, \qquad \forall g \in \mathrm{L}^2(Q) \,.$$

These assumptions were shown to be fairly general and we refer to Kanagawa et al. (2016, Section 4.1) for various examples where they are met. Next we state the main result of this section.

**Theorem 2.** *Let $\mathcal{X} = \mathbb{R}^d$, $\mathcal{Z} = \mathbb{R}^{d'}$, and $f : \mathcal{X} \to \mathcal{Z}$ be an $\alpha$-times differentiable function ($\alpha \in \mathbb{N}_+$). Take $s_1 > d/2$ and $s_2 > d'$ such that $s_1, s_2/2 \in \mathbb{N}_+$. Let $k_x^{s_1}$ and $k_z^{s_2}$ be Matérn kernels over $\mathcal{X}$ and $\mathcal{Z}$ respectively as defined in (2). Assume $X \sim P$ and $\hat{X} = \{(x_i, w_i)\}_{i=1}^n \sim S$ satisfy 1. Moreover, assume that $P$ and the marginals of $x_1, \ldots x_n$ have a common compact support. Suppose that, for some constants $b > 0$ and $0 < c \le 1/2$:*

*(i) $\mathbb{E}_S \left[ \|\hat{\mu}_X - \mu_X\|_{k_x^{s_1}}^2 \right] = O(n^{-2b})$ ;*
*(ii) $\sum_{i=1}^n w_i^2 = O(n^{-2c})$ (with probability 1) .*

*Let $\theta = \min(\frac{s_2}{2s_1}, \frac{\alpha}{s_1}, 1)$ and assume $\theta b - (1/2 - c)(1 - \theta) > 0$. Then*

$$\mathbb{E}_S \left[ \left\| \hat{\mu}_{f(X)} - \mu_{f(X)} \right\|_{k_z^{s_2}}^2 \right] = O \left( (\log n)^{d'} \, n^{-2\,(\theta b - (1/2 - c)(1 - \theta))} \right). \tag{3}$$

Before we provide a short sketch of the proof, let us briefly comment on this result. As a benchmark, remember that when $x_1, \ldots x_n$ are i.i.d. observations from $X$ and $\hat{X} = \{(x_i, 1/n)\}_{i=1}^n$, we get $\|\hat{\mu}_{f(X)} - \mu_{f(X)}\|^2 = O_P(n^{-1})$, which was recently shown to be a minimax optimal rate (Tolstikhin et al., 2016). How do we compare to this benchmark? In this case we have $b = c = 1/2$ and our rate is defined by $\theta$. If $f$ is smooth enough, say $\alpha > d/2 + 1$, and by setting $s_2 > 2s_1 = 2\alpha$, we recover the $O(n^{-1})$ rate up to an extra $(\log n)^{d'}$ factor.

However, Theorem 2 applies to much more general settings. Importantly, it makes no i.i.d. assumptions on the data points and weights, allowing for complex interdependences. Instead, it asks the convergence of the estimator $\hat{\mu}_X$ to the embedding $\mu_X$ to be sufficiently fast. On the downside, the upper bound is affected by the smoothness of $f$, even in the i.i.d. setting: if $\alpha \ll d/2$ the rate will become slower, as $\theta = \alpha/s_1$. Also, the rate depends both on $d$ and $d'$. Whether these are artefacts of our proof remains an open question.

*Proof.* Here we sketch the main ideas of the proof and develop the details in Appendix C. Throughout the proof, $C$ will designate a constant that depends neither on the sample size $n$ nor on the variable $R$ (to be introduced). $C$ may however change from line to line. We start by showing that:

$$\mathbb{E}_S \left[ \left\| \hat{\mu}_{f(X)}^{k_z} - \mu_{f(X)}^{k_z} \right\|_{k_z}^2 \right] = (2\pi)^{\frac{d'}{2}} \int_{\mathcal{Z}} \mathbb{E}_S \left[ \left( [\hat{\mu}_{f(X)}^h - \mu_{f(X)}^h](z) \right)^2 \right] dz, \tag{4}$$

where $h$ is Matérn kernel over $\mathcal{Z}$ with smoothness parameter $s_2/2$. Second, we upper bound the integrand by roughly imitating the proof idea of Theorem 1 from Kanagawa et al. (2016). This eventually yields:

$$\mathbb{E}_S \left[ \left( [\hat{\mu}_{f(X)}^h - \mu_{f(X)}^h](z) \right)^2 \right] \le C n^{-2\nu} , \tag{5}$$

where $\nu := \theta b - (1/2 - c)(1 - \theta)$. Unfortunately, this upper bound does not depend on $z$ and can not be integrated over the whole $\mathcal{Z}$ in (4). Denoting $B_R$ the ball of radius $R$, centred on the origin of $\mathcal{Z}$, we thus decompose the integral in (4) as:

$$\int_{\mathcal{Z}} \mathbb{E} \left[ \left( [\hat{\mu}_{f(X)}^h - \mu_{f(X)}^h](z) \right)^2 \right] dz$$
$$= \int_{B_R} \mathbb{E} \left[ \left( [\hat{\mu}_{f(X)}^h - \mu_{f(X)}^h](z) \right)^2 \right] dz + \int_{\mathcal{Z} \setminus B_R} \mathbb{E} \left[ \left( [\hat{\mu}_{f(X)}^h - \mu_{f(X)}^h](z) \right)^2 \right] dz.$$

On $B_R$ we upper bound the integral by (5) times the ball's volume (which grows like $R^d$):

$$\int_{B_R} \mathbb{E} \left[ \left( [\hat{\mu}_{f(X)}^h - \mu_{f(X)}^h](z) \right)^2 \right] dz \le C R^d n^{-2\nu} . \tag{6}$$

On $\mathcal{X} \setminus B_R$, we upper bound the integral by a value that decreases with $R$, which is of the form:

$$\int_{\mathcal{Z} \setminus B_R} \mathbb{E} \left[ \left( [\hat{\mu}_{f(X)}^h - \mu_{f(X)}^h](z) \right)^2 \right] dz \le C n^{1-2c} (R - C')^{s_2 - 2} e^{-2(R - C')} \tag{7}$$

with $C' > 0$ being a constant smaller than $R$. In essence, this upper bound decreases with $R$ because $[\hat{\mu}_{f(X)}^h - \mu_{f(X)}^h](z)$ decays with the same speed as $h$ when $\|z\|$ grows indefinitely. We are now left with two rates, (6) and (7), which respectively increase and decrease with growing $R$. We complete the proof by balancing these two terms, which results in setting $R \approx (\log n)^{1/2}$. $\qquad\square$

## 3 Functions of Multiple Arguments

The previous section applies to functions $f$ of one single variable $X$. However, we can apply its results to functions of multiple variables if we take the argument $X$ to be a tuple containing multiple values. In this section we discuss how to do it using two input variables from spaces $\mathcal{X}$ and $\mathcal{Y}$, but the results also apply to more inputs. To be precise, our input space changes from $\mathcal{X}$ to $\mathcal{X} \times \mathcal{Y}$, input random variable from $X$ to $(X, Y)$, and the kernel on the input space from $k_x$ to $k_{xy}$.

To apply our results from Section 2, all we need is a consistent estimator $\hat{\mu}_{(X,Y)}$ of the joint embedding $\mu_{(X,Y)}$. There are different ways to get such an estimator. One way is to sample $(x'_i, y'_i)$ i.i.d. from the joint distribution of $(X, Y)$ and construct the usual empirical estimator, or approximate it using reduced set methods. Alternatively, we may want to construct $\hat{\mu}_{(X,Y)}$ based only on consistent estimators of $\mu_X$ and $\mu_Y$. For example, this is how $\hat{\mu}_3$ was defined in Section 1.3. Below we show that this can indeed be done if $X$ and $Y$ are independent.

### 3.1 Application to Section 1.3

Following Schölkopf et al. (2015), we consider two independent random variables $X \sim P_x$ and $Y \sim P_y$. Their joint distribution is $P_x \otimes P_y$. Consistent estimators of their embeddings are given by $\hat{\mu}_X = \sum_{i=1}^n w_i k_x(x_i, .)$ and $\hat{\mu}_Y = \sum_{j=1}^n u_j k_y(y_i, .)$. In this section we show that $\hat{\mu}_{f(X,Y)} = \sum_{i,j=1}^n w_i u_j k_z\big(f(x_i, y_j), .\big)$ is a consistent estimator of $\mu_{f(X,Y)}$.

We choose a product kernel $k_{xy}\big((x_1, y_1), (x_2, y_2)\big) = k_x(x_1, x_2)k_y(y_1, y_2)$, so the corresponding RKHS is a tensor product $\mathcal{H}_{k_{xy}} = \mathcal{H}_{k_x} \otimes \mathcal{H}_{k_y}$ (Steinwart and Christmann, 2008, Lemma 4.6) and the mean embedding of the product random variable $(X, Y)$ is a tensor product of their marginal mean embeddings $\mu_{(X,Y)} = \mu_X \otimes \mu_Y$. With consistent estimators for the marginal embeddings we can estimate the joint embedding using their tensor product

$$\hat{\mu}_{(X,Y)} = \hat{\mu}_X \otimes \hat{\mu}_Y = \sum_{i,j=1}^n w_i u_j k_x(x_i, .) \otimes k_y(y_j, .) = \sum_{i,j=1}^n w_i u_j k_{xy}\big((x_i, y_j), (., .)\big).$$

If points are i.i.d. and $w_i = u_i = 1/n$, this reduces to the U-statistic estimator $\hat{\mu}_2$ from Section 1.3.

**Lemma 3.** *Let $(s_n)_n$ be any positive real sequence converging to zero. Suppose $k_{xy} = k_x k_y$ is a product kernel, $\mu_{(X,Y)} = \mu_X \otimes \mu_Y$, and $\hat{\mu}_{(X,Y)} = \hat{\mu}_X \otimes \hat{\mu}_Y$. Then:*

$$\begin{cases} \|\hat{\mu}_X - \mu_X\|_{k_x} = O(s_n); \\ \|\hat{\mu}_Y - \mu_Y\|_{k_y} = O(s_n) \end{cases} \quad \text{implies} \quad \left\|\hat{\mu}_{(X,Y)} - \mu_{(X,Y)}\right\|_{k_{xy}} = O(s_n).$$

*Proof.* For a detailed expansion of the first inequality see Appendix B.

$$\left\|\hat{\mu}_{(X,Y)} - \mu_{(X,Y)}\right\|_{k_{xy}} \le \|\mu_X\|_{k_x} \|\hat{\mu}_Y - \mu_Y\|_{k_y} + \|\mu_Y\|_{k_y} \|\hat{\mu}_X - \mu_X\|_{k_x}$$
$$+ \|\hat{\mu}_X - \mu_X\|_{k_x} \|\hat{\mu}_Y - \mu_Y\|_{k_y} = O(s_n) + O(s_n) + O(s_n^2) = O(s_n). \qquad\square$$

**Corollary 4.** *If $\hat{\mu}_X \xrightarrow[n\to\infty]{} \mu_X$ and $\hat{\mu}_Y \xrightarrow[n\to\infty]{} \mu_Y$, then $\hat{\mu}_{(X,Y)} \xrightarrow[n\to\infty]{} \mu_{(X,Y)}$.*

Together with the results from Section 2 this lets us reason about estimators resulting from applying functions to multiple independent random variables. Write

$$\hat{\mu}_{XY}^{k_{xy}} = \sum_{i,j=1}^n w_i u_j k_{xy}\big((x_i, y_j), .\big) = \sum_{\ell=1}^{n^2} \omega_\ell k_{xy}(\xi_\ell, .),$$

where $\ell$ enumerates the $(i,j)$ pairs and $\xi_\ell = (x_i, y_j)$, $\omega_\ell = w_i u_j$. Now if $\hat{\mu}_X^{k_x} \to \mu_X^{k_x}$ and $\hat{\mu}_Y^{k_y} \to \mu_Y^{k_y}$ then $\hat{\mu}_{XY}^{k_{xy}} \to \mu_{(X,Y)}^{k_{xy}}$ (according to Corollary 4) and Theorem 1 shows that $\sum_{i,j=1}^n w_i u_j k_z (f(x_i, y_j), .)$ is consistent as well. Unfortunately, we cannot apply Theorem 2 to get the speed of convergence, because a product of Matérn kernels is not a Matérn kernel any more.

One downside of this overall approach is that the number of expansion points used for the estimation of the joint increases exponentially with the number of arguments of $f$. This can lead to prohibitively large computational costs, especially if the result of such an operation is used as an input to another function of multiple arguments. To alleviate this problem, we may use reduced expansion set methods before or after applying $f$, as we did for example in Section 1.2.

To conclude this section, let us summarize the implications of our results for two practical scenarios that should be distinguished.

> ▷ If we have separate samples from two random variables $X$ and $Y$, then our results justify how to provide an estimate of the mean embedding of $f(X, Y)$ provided that $X$ and $Y$ are *independent*. The samples themselves need not be i.i.d. — we can also work with weighted samples computed, for instance, by a reduced set method.
> ▷ How about *dependent* random variables? For instance, imagine that $Y = -X$, and $f(X, Y) = X + Y$. Clearly, in this case the distribution of $f(X, Y)$ is a delta measure on 0, and there is no way to predict this from separate samples of $X$ and $Y$. However, it should be stressed that our results (consistency and finite sample bound) apply even to the case where $X$ and $Y$ are dependent. In that case, however, they require a consistent estimator of the joint embedding $\mu_{(X,Y)}$.
> ▷ It is also sufficient to have a reduced set expansion of the embedding of the joint distribution. This setting may sound strange, but it potentially has significant applications. Imagine that one has a large database of user data, sampled from a joint distribution. If we expand the joint's embedding in terms of *synthetic* expansion points using a reduced set construction method, then we can pass on these (weighted) synthetic expansion points to a third party without revealing the original data. Using our results, the third party can nevertheless perform arbitrary continuous functional operations on the joint distribution in a consistent manner.

## 4   Conclusion and future work

This paper provides a theoretical foundation for using kernel mean embeddings as approximate representations of random variables in scenarios where we need to apply functions to those random variables. We show that for continuous functions $f$ (including all functions on discrete domains), consistency of the mean embedding estimator of a random variable $X$ implies consistency of the mean embedding estimator of $f(X)$. Furthermore, if the kernels are Matérn and the function $f$ is sufficiently smooth, we provide bounds on the convergence rate. Importantly, our results apply beyond i.i.d. samples and cover estimators based on expansions with interdependent points and weights. One interesting future direction is to improve the finite-sample bounds and extend them to general radial and/or translation-invariant kernels.

Our work is motivated by the field of probabilistic programming. Using our theoretical results, kernel mean embeddings can be used to generalize functional operations (which lie at the core of all programming languages) to distributions over data types in a principled manner, by applying the operations to the points or approximate kernel expansions. This is in principle feasible for any data type provided a suitable kernel function can be defined on it. We believe that the approach holds significant potential for future probabilistic programming systems.

**Acknowledgements**

We thank Krikamol Muandet for providing the code used to generate Figure 1, Paul Rubenstein, Motonobu Kanagawa and Bharath Sriperumbudur for very useful discussions, and our anonymous reviewers for their valuable feedback. Carl-Johann Simon-Gabriel is supported by a Google European Fellowship in Causal Inference.

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
