[Supplementary Material · ssts16nips_supp.pdf]

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

## Footnotes

[1]Adams and Fournier (2003) introduce interpolation spaces using so-called *J*- and *K-methods*, resulting in two notations $(E_0, E_1)_{\theta,q;J}$ (Definition 7.12) and $(E_0, E_1)_{\theta,q;K}$ (Definition 7.9) respectively. However, it follows from Theorem 7.16 that these two definitions are equivalent if $0 < \theta < 1$ and we simply drop the $K$ and $J$ subindices.

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

# A Detailed Proof of Theorem 1

*Proof.*

$$\left\|\hat{\mu}_Q^{k_z} - \mu_Q^{k_z}\right\|_{k_z}^2 = \left\|\sum_{i=1}^n w_i k_z(f(x_i),.) - \mathbb{E}\left[k_z(f(X),.)\right]\right\|_{k_z}^2$$

$$= \langle \sum_{i=1}^n w_i k_z(f(x_i),.) - \mathbb{E}\left[k_z(f(X),.)\right], \sum_{j=1}^n w_j k_z(f(x_j),.) - \mathbb{E}\left[k_z(f(X'),.)\right]\rangle$$

$$= \sum_{i,j=1}^n w_i w_j \langle k_z(f(x_i),.), k_z(f(x_j),.)\rangle - 2\sum_{i=1}^n w_i \mathbb{E}\left[\langle k_z(f(x_i),.), k_z(f(X),.)\rangle\right] + \mathbb{E}\left[\langle k_z(f(X),.), k_z(f(X'),.)\rangle\right]$$

$$= \sum_{i,j=1}^n w_i w_j k_z\big(f(x_i), f(x_j)\big) - 2\sum_{i=1}^n w_i \mathbb{E}\left[k_z\big(f(x_i), f(X)\big)\right] + \mathbb{E}\left[k_z(f(X), f(X'))\right]$$

$$= \sum_{i,j=1}^n w_i w_j \widetilde{k}_x(x_i, x_j) - 2\sum_{i=1}^n w_i \mathbb{E}\left[\widetilde{k}_x(x_i, X)\right] + \mathbb{E}\left[\widetilde{k}_x(X, X')\right]$$

$$= \sum_{i,j=1}^n w_i w_j \langle \widetilde{k}_x(x_i,.), \widetilde{k}_x(x_j,.)\rangle - 2\sum_{i=1}^n w_i \mathbb{E}\left[\langle \widetilde{k}_x(x_i,.), \widetilde{k}_x(X,.)\rangle\right] + \mathbb{E}\left[\langle \widetilde{k}_x(X,.), \widetilde{k}_x(X',.)\rangle\right]$$

$$= \langle \sum_{i=1}^n w_i \widetilde{k}_x(x_i,.) - \mathbb{E}\left[\widetilde{k}_x(X,.)\right], \sum_{j=1}^n w_j \widetilde{k}_x(x_j,.) - \mathbb{E}\left[\widetilde{k}_x(X',.)\right]\rangle$$

$$= \left\|\sum_{i=1}^n w_i \widetilde{k}_x(x_i,.) - \mathbb{E}\left[\widetilde{k}_x(X,.)\right]\right\|_{\widetilde{k}_x}^2 = \left\|\hat{\mu}_X^{\widetilde{k}_x} - \mu_X^{\widetilde{k}_x}\right\|_{\widetilde{k}_x}^2 \xrightarrow[n\to\infty]{} 0. \qquad \square$$

# B Detailed Proof of Lemma 3

*Proof.*

$$\left\|\hat{\mu}_{XY}^{k_{xy}} - \mu_{XY}^{k_{xy}}\right\|_{k_{xy}} = \left\|\hat{\mu}_X^{k_x} \otimes \hat{\mu}_Y^{k_y} - \mu_X^{k_x} \otimes \mu_Y^{k_y}\right\|_{k_{xy}}$$

$$= \left\|\hat{\mu}_X^{k_x} \otimes \hat{\mu}_Y^{k_y} - \hat{\mu}_X^{k_x} \otimes \mu_Y^{k_y} + \hat{\mu}_X^{k_x} \otimes \mu_Y^{k_y} - \mu_X^{k_x} \otimes \mu_Y^{k_y}\right\|_{k_{xy}}$$

$$= \left\|\hat{\mu}_X^{k_x} \otimes (\hat{\mu}_Y^{k_y} - \mu_Y^{k_y}) + (\hat{\mu}_X^{k_x} - \mu_X^{k_x}) \otimes \mu_Y^{k_y}\right\|_{k_{xy}}$$

$$\leq \left\|\hat{\mu}_X^{k_x}\right\|_{k_x} \left\|\hat{\mu}_Y^{k_y} - \mu_Y^{k_y}\right\|_{k_y} + \left\|\mu_Y^{k_y}\right\|_{k_y} \left\|\hat{\mu}_X^{k_x} - \mu_X^{k_x}\right\|_{k_x}$$

$$= \left\|\mu_X^{k_x} + \hat{\mu}_X^{k_x} - \mu_X^{k_x}\right\|_{k_x} \left\|\hat{\mu}_Y^{k_y} - \mu_Y^{k_y}\right\|_{k_y} + \left\|\mu_Y^{k_y}\right\|_{k_y} \left\|\hat{\mu}_X^{k_x} - \mu_X^{k_x}\right\|_{k_x}$$

$$\leq \left\|\mu_X^{k_x}\right\|_{k_x} \left\|\hat{\mu}_Y^{k_y} - \mu_Y^{k_y}\right\|_{k_y} + \left\|\mu_Y^{k_y}\right\|_{k_y} \left\|\hat{\mu}_X^{k_x} - \mu_X^{k_x}\right\|_{k_x}$$

$$\quad + \left\|\hat{\mu}_X^{k_x} - \mu_X^{k_x}\right\|_{k_x} \left\|\hat{\mu}_Y^{k_y} - \mu_Y^{k_y}\right\|_{k_y}$$

$$= O(s_n) + O(s_n) + O(s_n^2) = O(s_n + s_n^2). \qquad \square$$

# C Detailed Proof of Theorem 2

## C.1 Notations, Reminders and Preliminaries

For any function $\psi \in \mathrm{L}^1(\mathbb{R}^d)$ and any finite (signed or complex regular Borel) measure $\nu$ over $\mathbb{R}^d$, we define their convolution as:

$$\nu * \psi(x) := \int \psi(x - x') \, d\nu(x') .$$

We define the Fourier and inverse Fourier transforms of $\psi$ and $\nu$ as

$$\mathscr{F}\,\psi(\omega) := (2\pi)^{-d/2} \int_{\mathbb{R}^d} e^{-i\langle \omega, x\rangle} \psi(x)\, dx \quad \text{and} \quad \mathscr{F}\,\nu(\omega) := (2\pi)^{-d/2} \int_{\mathbb{R}^d} e^{-i\langle \omega, x\rangle}\, d\nu(x)\,,$$

$$\mathscr{F}^{-1}\,\psi(\omega) := (2\pi)^{-d/2} \int_{\mathbb{R}^d} e^{i\langle \omega, x\rangle} \psi(x)\, dx \quad \text{and} \quad \mathscr{F}^{-1}\,\nu(\omega) := (2\pi)^{-d/2} \int_{\mathbb{R}^d} e^{i\langle \omega, x\rangle}\, d\nu(x)\,.$$

Fourier transforms are of particular interest when working with translation-invariant kernel because of Bochner's theorem. Here we quote Wendland (2004, Theorem 6.6), but add a useful second sentence, which is immediate to show.

**Theorem 5** (Bochner). *A continuous function $\psi : \mathbb{R}^d \to \mathbb{C}$ is positive definite if and only if it is the Fourier transform of a finite, nonnegative Borel measure $\nu$ over $\mathbb{R}^d$. Moreover, $\psi$ is real-valued if and only if $\nu$ is symmetric.*

The next theorem, also quoted from Wendland (2004, Corollary 5.25), shows that the Fourier (inverse) transform may be seen as a unitary isomorphism from $\mathrm{L}^2(\mathbb{R}^d)$ to $\mathrm{L}^2(\mathbb{R}^d)$.

**Theorem 6** (Plancherel). *There exists an isomorphic mapping $T : \mathrm{L}^2(\mathbb{R}^d) \to \mathrm{L}^2(\mathbb{R}^d)$ such that:*

*(i) $\|Tf\|_{\mathrm{L}^2(\mathbb{R}^d)} = \|f\|_{\mathrm{L}^2(\mathbb{R}^d)}$ for all $f \in \mathrm{L}^2(\mathbb{R}^d)$.*
*(ii) $Tf = \mathscr{F}\,f$ for any $f \in \mathrm{L}^2(\mathbb{R}^d) \cap \mathrm{L}^1(\mathbb{R}^d)$.*
*(iii) $T^{-1}g = \mathscr{F}^{-1}\,g$ for all $g \in \mathrm{L}^2(\mathbb{R}^d) \cap \mathrm{L}^1(\mathbb{R}^d)$.*

*The isomorphism is uniquely determined by these properties.*

We will call $T$ the Fourier transform over $\mathrm{L}^2$ and note it $\mathscr{F}$.

*Remark* 7. Combining Plancherel's and Bochner's theorems, we see that, if $\psi$ is a continuous, positive definite (resp. and real-valued) function in $\mathrm{L}^2(\mathbb{R}^d)$, then the measure $\nu$ from Bochner's theorem is absolutely continuous, and its density is $\mathscr{F}^{-1}\,\psi$. In particular, $\mathscr{F}^{-1}\,\psi$ is real-valued, nonnegative (resp. and symmetric).

Next, our proof of Theorem 2 will need the following result.

**Lemma 8.** *Let $\mathcal{Z} = \mathbb{R}^{d'}$, $\psi \in \mathrm{L}^2(\mathbb{R}^d)$ such that $\mathscr{F}\,\psi \in \mathrm{L}^1(\mathbb{R}^d)$. Let $k$ be the translation-invariant kernel $k(z, z') := \psi(z - z')$ and $h(z) := \mathscr{F}^{-1}\sqrt{\mathscr{F}\,\psi}(z)$. Let $Z$ be any random variable on $\mathcal{Z}$, and $\hat{Z} := \{(z_i, w_i)\}_{i=1}^n$. Then:*

$$\left\|\hat{\mu}_Z^k - \mu_Z^k\right\|_k^2 = (2\pi)^{\frac{d'}{2}} \int_{z \in \mathcal{Z}} \left|\hat{\mu}_Z^h - \mu_Z^h\right|^2 dz\,. \tag{8}$$

*Proof.* (of Lemma 8) For any finite (signed) measure $\nu$ over $\mathcal{Z} = \mathbb{R}^{d'}$, we define:

$$\mu_\nu^k := \int k(z, .)\, d\nu(z)\,.$$

Then we have:

$$\begin{aligned}
\left\|\mu_\nu^k\right\|_k^2 &= \int_{z \in \mathbb{R}^d} \int_{z' \in \mathbb{R}^d} \psi(z - z')\, d\nu(z)\, d\nu(z') \\
&= \int_{z \in \mathbb{R}^d} \int_{z' \in \mathbb{R}^d} \left((2\pi)^{-d'/2} \int_{\omega \in \mathbb{R}^d} e^{-i\langle \omega, z - z'\rangle}\, \mathscr{F}^{-1}\,\psi(\omega)\, d\omega\right) d\nu(z)\, d\nu(z') \\
&= \int_{\omega \in \mathbb{R}^d} (2\pi)^{-d'/2} \int_{z \in \mathbb{R}^d} \int_{z' \in \mathbb{R}^d} e^{-i\langle \omega, z - z'\rangle}\, d\nu(z)\, d\nu(z')\, \mathscr{F}^{-1}\,\psi(\omega)\, d\omega \\
&= \int_{\omega \in \mathbb{R}^d} (2\pi)^{d'/2}\, \mathscr{F}\,\nu(\omega)\, \mathscr{F}\,\nu(-\omega)\, \mathscr{F}^{-1}\,\psi(\omega)\, d\omega \\
&= (2\pi)^{d'/2} \int_{\omega \in \mathbb{R}^d} |\mathscr{F}\,\nu(\omega)|^2\, \mathscr{F}^{-1}\,\psi(\omega)\, d\omega
\end{aligned}$$

Second line uses the following: (i) $\psi$ is continuous, because $\mathscr{F}\,\psi \in \mathrm{L}^1(\mathbb{R}^d)$ (Riemann-Lebesgue lemma); (ii) Theorem 5 (Bochner) and Remark 7 from the Appendix. Third and fourth line use

Fubini's theorem. Last line uses the fact that $\mathscr{F}\,\nu(-\omega)$ is the complex conjugate of $\mathscr{F}\,\nu$ because $\mathscr{F}\,\psi$ is positive (thus real-valued).

Applying this with $\nu = \hat{Q} - Q$, where $Q$ is the distribution of $Z$ and $\hat{Q} := \sum_i w_i \delta_{z_i}$, we get:

$$
\begin{aligned}
\left\| \hat{\mu}_Z^k - \mu_Z^k \right\|_k^2 &= \left\| \mu_{\hat{Q}-Q}^k \right\|_k^2 \\
&= (2\pi)^{d'/2} \int_{\omega \in \mathbb{R}^d} \left| \mathscr{F}[\hat{Q}-Q](\omega) \right|^2 \mathscr{F}\,\psi(\omega)\,d\omega \\
&= (2\pi)^{d'/2} \int_{\omega \in \mathbb{R}^d} \left| \mathscr{F}[\hat{Q}-Q](\omega) \sqrt{\mathscr{F}\,\psi(\omega)} \right|^2 d\omega \\
&= (2\pi)^{d'/2} \int_{z \in \mathcal{Z}} \left| \mathscr{F}^{-1}\left[ \mathscr{F}[\hat{Q}-Q]\sqrt{\mathscr{F}\,\psi} \right](z) \right|^2 dz \\
&= (2\pi)^{d'/2} \int_{z \in \mathcal{Z}} \left| [\hat{Q}-Q] * h(z) \right|^2 dz \\
&= (2\pi)^{d'/2} \int_{z \in \mathcal{Z}} \left| \sum_i w_i h(z - z_i) - \int h(z-s)\,dQ(s) \right|^2 dz \\
&= (2\pi)^{d'/2} \int_{z \in \mathcal{Z}} \left| \hat{\mu}_Z^h - \mu_Z^h(z) \right|^2 dz \,.
\end{aligned}
$$

Third line uses the fact that $\mathscr{F}\,\psi$ is positive (see Appendix, Remark 7). Fourth line uses Plancherel's theorem (see Appendix, Theorem 6). Fifth line uses the fact that the Fourier (inverse) transform of a product equals the convolutional product of the (inverse) Fourier transforms (Katznelson, 2004, Theorem 1.4, and its generalisation to finite measures p.145). □

We now state Theorem 1 from Kanagawa et al. (2016), which serves as basis to our proof. Slightly modifying[1] the notation of Adams and Fournier (2003, Chapter 7), for $0 < \theta < 1$ and $1 \leq q \leq \infty$ we will write $(E_0, E_1)_{\theta,q}$ to denote interpolation spaces, where $E_0$ and $E_1$ are Banach spaces that are continuously embedded into some topological Hausdorff vector space $\mathcal{E}$. Following Kanagawa et al. (2016), we also define $(E_0, E_1)_{1,2} := E_1$.

**Theorem 9** (Kanagawa et al.). *Let $X$ be a random variable with distribution $P$ and let $\{(x_i, w_i)\}_{i=1}^n$ be random variables with joint distribution $S$ satisfying Assumption 1 (with corresponding distribution $Q$). Let $\hat{\mu}_X := \sum_i w_i k(x_i, .)$ be an estimator of $\mu_X := \int k(x, .)\,dP(x)$ such that for some constants $b > 0$ and $0 < c \leq 1/2$:*

*(i) $\mathbb{E}_S\left[ \|\hat{\mu}_X - \mu_X\|_k \right] = O(n^{-b})$,*
*(ii) $\mathbb{E}_S\left[ \sum_i w_i^2 \right] = O(n^{-2c})$*

*as $n \to \infty$. Let $\theta$ be a constant such that $0 < \theta \leq 1$.*

*Then, for any function $g : \mathbb{R}^d \to \mathbb{R}$ in $\left( \mathrm{L}^2(Q), \mathcal{H}_k \right)_{\theta,2}$, there exists a constant $C$, independent of $n$, such that:*

$$
\mathbb{E}_S\left[ \left| \sum_i w_i g(x_i) - \mathbb{E}_{X \sim P}[g(X)] \right| \right] \leq C\, n^{-\theta b + (1/2 - c)(1 - \theta)} \,. \tag{9}
$$

In the proof of our finite sample guarantee, we will need the following slightly modified version of this result, where we (a) slightly modify condition (ii) by asking that it holds almost surely, and (b) consider squared norms in Condition (i) and (9).

**Theorem 10** (Kanagawa et al.). *Let $X$ be a random variable with distribution $P$ and let $\{(x_i, w_i)\}$ be random variables with joint distribution $S$ satisfying Assumption 1 (with corresponding distribution $Q$). Let $\hat{\mu}_X := \sum_i w_i k(x_i, .)$ be an estimator of $\mu_X := \int k(x, .)\,dP(x)$ such that for some constants $b > 0$ and $0 < c \leq 1/2$:*

(i) $\mathbb{E}_S\left[\|\hat{\mu}_X - \mu_X\|_k^2\right] = O(n^{-2b})$,

(ii) $\sum_{i=1}^n w_i^2 = O(n^{-2c})$ *(with S-probability 1)* ,

*as $n \to \infty$. Let $\theta$ be a constant such that $0 < \theta \le 1$.*

*Then, for any function $g : \mathbb{R}^d \to \mathbb{R}$ in $\left(\mathrm{L}^2(Q), \mathcal{H}_k\right)_{\theta,2}$, there exists a constant $C$, independent of $n$, such that:*

$$\mathbb{E}_S\left[\left|\sum_i w_i g(x_i) - \mathbb{E}_{X\sim P}[g(X)]\right|\right] \le C\, n^{-2\left(\theta b - (1/2 - c)(1-\theta)\right)} . \tag{10}$$

*Proof.* The proof of this adapted version of Kanagawa et al. (2016, Theorem 1) is almost a copy paste of the original proof, but with the appropriate squares to account for the modified condition (i), and with their $f$ renamed to $g$ here. The only slight non-trivial difference is in their Inequality (20). Replace their triangular inequality by Jensen's inequality to yield:

$$\mathbb{E}_S\left[\left|\sum_{i=1}^n w_i g(x_i) - \mathbb{E}_{X\sim P}[g(X)]\right|^2\right] \le 3\,\mathbb{E}_S\left[\left|\sum_{i=1}^n w_i g(x_i) - \sum_{i=1}^n w_i g_{\lambda_n}(x_i)\right|^2\right]$$

$$+ 3\,\mathbb{E}_S\left[\left|\sum_{i=1}^n w_i g_{\lambda_n}(x_i) - \mathbb{E}_{X\sim P}[g_{\lambda_n}(X)]\right|^2\right]$$

$$+ 3\,\mathbb{E}_S\left[\left|\mathbb{E}_{X\sim P}[g_{\lambda_n}(X)] - \mathbb{E}_{X\sim P}[g(X)]\right|^2\right] ,$$

where $g$ and $g_{\lambda_n}$ are the functions that they call $f$ and $f_{\lambda_n}$. $\qquad\square$

We are now ready to prove Theorem 2.

## C.2   Proof of Theorem 2

*Proof.* This proof is self-contained: the sketch from the main part is not needed. Throughout the proof, $C$ designates constants that depend neither on sample size $n$ nor on radius $R$ (to be introduced). But their value may change from line to line.

Let $\psi$ be such that $k_z^{s_2}(z, z') = \psi(z - z')$. Then $\mathscr{F}\psi(\omega) = (1 + \|\omega\|_2^2)^{-s_2}$ (Wendland, 2004, Chapter 10). Applying Lemma 8 to the Matérn kernel $k_z^{s_2}$ thus yields:

$$\mathbb{E}_S\left[\left\|\hat{\mu}_{f(X)}^{k_z^{s_2}} - \mu_{f(X)}^{k_z^{s_2}}\right\|_{k_z^{s_2}}^2\right] = (2\pi)^{\frac{d'}{2}} \int_{\mathcal{Z}} \mathbb{E}_S\left[\left([\hat{\mu}_{f(X)}^h - \mu_{f(X)}^h](z)\right)^2\right] dz, \tag{11}$$

where $h = \mathscr{F}^{-1}\sqrt{\mathscr{F}k_z^{s_2}}$ is again a Matérn kernel, but with smoothness parameter $s_2/2 > d'/2$.

### Step 1: Applying Theorem 10

We now want to upper bound the integrand by using Theorem 10. To do so, let $\mathcal{K}$ be the common compact support of $P$ and marginals of $x_1, \ldots, x_n$. Now, rewrite the integrand as:

$$\mathbb{E}_S\left[\left([\hat{\mu}_{f(X)}^h - \mu_{f(X)}^h](z)\right)^2\right] = \mathbb{E}_S\left[\left(\sum_i w_i h\big(f(x_i) - z\big) - \mathbb{E}_{X\sim P}\left[h\big(f(X) - z\big)\right]\right)^2\right]$$

$$= \mathbb{E}_S\left[\left(\sum_i w_i h\big(f(x_i) - z\big)\varphi_\mathcal{K}(x_i) - \mathbb{E}_{X\sim P}\left[h\big(f(X) - z\big)\varphi_\mathcal{K}(X)\right]\right)^2\right]$$

$$= \mathbb{E}_S\left[\left(\sum_i w_i g_z(x_i) - \mathbb{E}_{X\sim P}[g_z(X)]\right)^2\right] , \tag{12}$$

where $\varphi_\mathcal{K}$ is any smooth function $\leq 1$, with compact support, that equals 1 on a neighbourhood of $\mathcal{K}$ and where $g_z(x) := h(f(x) - z)\varphi_\mathcal{K}(x)$.

To apply Theorem 10, we need to prove the existence of $0 < \theta \leq 1$ such that $g_z \in \left(L^2(Q), \mathcal{H}_{k_x^{s_1}}\right)_{\theta,2}$ for each $z \in \mathcal{Z}$. We will prove this fact in two steps: (a) first we show that $g_z \in \left(L^2(\mathbb{R}^d), \mathcal{H}_{k_x^{s_1}}\right)_{\theta,2}$ for each $z \in \mathcal{Z}$ and certain choice of $\theta$ and (b) we argue that $\left(L^2(\mathbb{R}^d), \mathcal{H}_{k_x^{s_1}}\right)_{\theta,2}$ is continuously embedded in $\left(L^2(Q), \mathcal{H}_{k_x^{s_1}}\right)_{\theta,2}$.

**Step (a):** Note that $g_z \in \mathcal{W}_2^{\min(\alpha, s_2/2)}(\mathbb{R}^d)$ because $f$ is $\alpha$-times differentiable, $h \in \mathcal{W}_2^{s_2/2}(\mathbb{R}^{d'})$ (thus $g_z$ is $\min(\alpha, s_2/2)$-times differentiable in the distributional sense), and $g_z$ has compact support (thus meets the integrability conditions of Sobolev spaces). As $k_x^{s_1}$ is a Matérn kernel with smoothness parameter $s_1$, its associated RKHS $\mathcal{H}_{k_x^{s_1}}$ is the Sobolev space $\mathcal{W}_2^{s_1}(\mathbb{R}^d)$ (Wendland, 2004, Chapter 10). Now, if $s_1 \leq \min(\alpha, s_2/2)$, then $g_z \in \mathcal{W}_2^{s_1}(\mathbb{R}^d) = \mathcal{H}_{k_x^{s_1}} = \left(L^2(\mathbb{R}^d), \mathcal{W}_2^{s_1}(\mathbb{R}^d)\right)_{1,2}$ and step (a) holds for $\theta = 1$. Thus for the rest of this step, we assume $s_1 > \min(\alpha, s_2/2)$. It is known that $\mathcal{W}_2^s(\mathbb{R}^d) = B_{2,2}^s(\mathbb{R}^d)$ for $0 < s < \infty$ (Adams and Fournier, 2003, Page 255), where $B_{2,2}^s(\mathbb{R}^d)$ is the Besov space of smoothness $s$. It is also known that $B_{2,2}^s(\mathbb{R}^d) = \left(L^2(\mathbb{R}^d), \mathcal{W}_2^m(\mathbb{R}^d)\right)_{s/m,2}$ for any integer $m > s$ (Adams and Fournier, 2003, Page 230). Applying this to $\mathcal{W}_2^{\min(\alpha, s_2/2)}(\mathbb{R}^d)$ and denoting $s' = \min(\alpha, s_2/2)$ we get

$$g_z \in \mathcal{W}_2^{s'}(\mathbb{R}^d) = \left(L^2(\mathbb{R}^d), \mathcal{W}_2^{s_1}(\mathbb{R}^d)\right)_{s'/s_1,2} = \left(L^2(\mathbb{R}^d), \mathcal{H}_{k_x^{s_1}}\right)_{s'/s_1,2}, \qquad \forall z \in \mathcal{Z} \ .$$

Thus, whatever $s_1$, step (a) is always satisfied with $\theta := \min(\frac{\alpha}{s_1}, \frac{s_2}{2s_1}, 1) \leq 1$.

**Step (b):** If $\theta = 1$, then $\left(L^2(\mathbb{R}^d), \mathcal{H}_{k_x^{s_1}}\right)_{1,2} = \mathcal{H}_{k_x^{s_1}} = \left(L^2(Q), \mathcal{H}_{k_x^{s_1}}\right)_{1,2}$. Now assume $\theta < 1$. Note that $L^2(\mathbb{R}^d)$ is continuously embedded in $L^2(Q)$, because we assumed that $Q$ has a bounded density. Thus Theorem V.1.12 of Bennett and Sharpley (1988) applies and gives the desired inclusion.

Now we apply Theorem 10, which yields a constant $C_z$ independent of $n$ such that:

$$\mathbb{E}_S\left[\left([\hat{\mu}_{f(X)}^h - \mu_{f(X)}^h](z)\right)^2\right] \leq C_z n^{-2\nu} \ ,$$

with $\nu := \theta b - (1/2 - c)(1 - \theta)$.

We now prove that the constants $C_z$ are uniformly bounded. From Equations (18-19) of Kanagawa et al. (2016), it appears that $C_z = C \left\| T^{-\theta/2} g_z \right\|_{L^2(Q)}$, where $C$ is a constant independent of $z$ and $T^{-\theta/2}$ is defined as follows. Let $T$ be the operator from $L^2(Q)$ to $L^2(Q)$ defined by

$$Tf := \int k_x(x, .) f(x) \, dQ(x).$$

It is continuous, compact and self-adjoint. Denoting $(e_i)_i$ an orthonormal basis of eigenfunctions in $L^2(Q)$ with eigenvalues $\mu_1 \geq \mu_2 \geq \cdots \geq 0$, let $T^{\theta/2}$ be the operator from $L^2(Q)$ to $L^2(Q)$ defined by:

$$T^{\theta/2}f := \sum_{i=1}^{\infty} \mu_i^{\theta/2} \langle e_i, f \rangle_{L^2(Q)} e_i \ .$$

Using Scovel et al. (2014, Corollary 4.9.i) together with Steinwart and Christmann (2008, Theorem 4.26.i) we conclude that $T^{\theta/2}$ is injective. Thus $\mu_i > 0$ for all $i$. Thus, if $\theta = 1$, Lemma 6.4 of Steinwart and Scovel (2012) shows that the range of $T^{\theta/2}$ is $[\mathcal{H}_k]_\sim$, the image of the canonical embedding of $\mathcal{H}_k$ into $L^2(Q)$. And as $Q$ has full support, we may identify $[\mathcal{H}_k]_\sim$ and $\mathcal{H}_k = \left(L^2(Q), \mathcal{H}_k\right)_{\theta,2}$. Now, if $\theta < 1$, Theorem 4.6 of Steinwart and Scovel (2012) shows that the range of $T^{\theta/2}$ is $\left(L^2(Q), \mathcal{H}_k\right)_{\theta,2}$.

Thus the inverse operator $T^{-\theta/2}$ is well-defined, goes from $\left(L^2(Q), \mathcal{H}_k\right)_{\theta,2}$ to $L^2(Q)$ and can be written in the following form:

$$T^{-\theta/2}f := \sum_{i=1}^{\infty} \mu_i^{-\theta/2} \langle e_i, f \rangle_{L^2(Q)} e_i \ . \tag{13}$$

Using this, we get:

$$|C_z| = C \left\| T^{-\theta/2} g_z \right\|_{\mathrm{L}^2(Q)}$$

$$= C \left\| \sum_{i=1}^{\infty} \mu_i^{-\theta/2} \langle e_i, h(f(\cdot) - z)\varphi_{\mathcal{K}}(\cdot)\rangle_{\mathrm{L}^2(Q)} e_i \right\|_{\mathrm{L}^2(Q)}$$

$$\leq C \max_{z \in \mathcal{Z}} |h(z)| \left\| \sum_{i=1}^{\infty} \mu_i^{-\theta/2} \langle e_i, \varphi_{\mathcal{K}}\rangle_{\mathrm{L}^2(Q)} e_i \right\|_{\mathrm{L}^2(Q)}$$

$$= C \max_{z \in \mathcal{Z}} |h(z)| \left\| T^{-\theta/2} \varphi_{\mathcal{K}} \right\|_{\mathrm{L}^2(Q)} ,$$

which is a constant independent of $z$. Hereby, we used the fact that $\varphi_{\mathcal{K}} \in \left(\mathrm{L}^2(Q), \mathcal{H}_k\right)_{\theta,2}$, because it is infinitely smooth and has compact support. Thus we just proved that

$$\underset{S}{\mathbb{E}} \left[ \left( [\hat{\mu}_{f(X)}^h - \mu_{f(X)}^h](z) \right)^2 \right] \leq C n^{-2\nu} . \tag{14}$$

**Step 2: Splitting the integral in two parts**

However, now that this upper bound does not depend on $z$ anymore, we cannot integrate over all $\mathcal{Z}$ $(= \mathbb{R}^{d'})$. Thus we now decompose the integral in (11) as:

$$\int_{\mathcal{Z}} \underset{S}{\mathbb{E}} \left[ \left( [\hat{\mu}_{f(X)}^h - \mu_{f(X)}^h](z) \right)^2 \right] dz$$

$$= \int_{B_R} \underset{S}{\mathbb{E}} \left[ \left( [\hat{\mu}_{f(X)}^h - \mu_{f(X)}^h](z) \right)^2 \right] dz + \int_{\mathcal{Z} \backslash B_R} \underset{S}{\mathbb{E}} \left[ \left( [\hat{\mu}_{f(X)}^h - \mu_{f(X)}^h](z) \right)^2 \right] dz , \tag{15}$$

where $B_R$ denotes the ball of radius $R$, centred on the origin of $\mathcal{Z} = \mathbb{R}^{d'}$. We will upper bound each term by a function depending on $R$, and eventually make $R$ depend on the sample size so as to balance both upper bounds.

On $B_R$ we upper bound the integral by Rate (14) times the ball's volume (which grows like $R^{d'}$):

$$\int_{B_R} \underset{S}{\mathbb{E}} \left[ \left( [\hat{\mu}_{f(X)}^h - \mu_{f(X)}^h](z) \right)^2 \right] dz \leq C R^{d'} n^{-2\nu} . \tag{16}$$

On $\mathcal{Z} \backslash B_R$ we upper bound the integral by a value that decreases with $R$. The intuition is that, according to (12), the integrand is the expectation of sums of Matérn functions, which are all centred on a compact domain. Thus it should decay exponentially with $z$ outside of a sufficiently large ball. Next we turn to the formal argument.

Let us define $\|f\|_{\mathcal{K}} := \max_{x \in \mathcal{X}} \|f(x)\varphi_{\mathcal{K}}(x)\|$, which is finite because $f\varphi_{\mathcal{K}}$ is an $\alpha$-times differentiable (thus continuous) function with compact support. Now, Matérn kernels are radial kernels, meaning that there exists a function $\tilde{h}$ over $\mathbb{R}$ such that $h(x) = \tilde{h}(\|x\|)$ (Tolstikhin et al., 2016, page 5). Moreover $\tilde{h}$ is strictly positive and decreasing. Using (12) we may write

$$\underset{S}{\mathbb{E}} \left[ \left( [\hat{\mu}_{f(X)}^h - \mu_{f(X)}^h](z) \right)^2 \right]$$

$$= \underset{S}{\mathbb{E}} \left[ \left( \sum_i w_i h\big(f(x_i) - z\big)\varphi_{\mathcal{K}}(x_i) - \underset{X \sim P}{\mathbb{E}} \big[ h\big(f(X) - z\big)\varphi_{\mathcal{K}}(X)\big] \right)^2 \right]$$

$$\leq \underset{S}{\mathbb{E}} \left[ \left( \sum_i w_i h\big(f(x_i) - z\big)\varphi_{\mathcal{K}}(x_i) \right)^2 + \left( \underset{X \sim P}{\mathbb{E}} \big[ h\big(f(X) - z\big)\varphi_{\mathcal{K}}(X)\big] \right)^2 \right]$$

$$\overset{(\dagger)}{\leq} \tilde{h}(\|z\| - \|f\|_{\mathcal{K}})^2 \, \underset{S}{\mathbb{E}} \left[ \left( \left( \sum_i w_i \right)^2 + 1 \right) \right] ,$$

where we assumed $R > \|f\|_{\mathcal{K}}$ and used the fact that $\tilde{h}$ is a decreasing function in (†). Using Cauchy-Schwarz and applying hypothesis (ii), we get:

$$\mathbb{E}\left[\left([\hat{\mu}^h_{f(X)} - \mu^h_{f(X)}](z)\right)^2\right] \leq \tilde{h}(\|z\| - \|f\|_{\mathcal{K}})^2 \underset{S}{\mathbb{E}}\left[\left(n\left(\sum_i w_i^2\right) + 1\right)\right]$$

$$\leq C\, n^{1-2c}\, \tilde{h}(\|z\| - \|f\|_{\mathcal{K}})^2 \, .$$

Let $\mathcal{S}_{d'}$ be the surface area of the unit sphere in $\mathbb{R}^{d'}$. We have:

$$\int_{\mathcal{Z}\setminus B_R} \mathbb{E}\left[\left([\hat{\mu}^h_{f(X)} - \mu^h_{f(X)}](z)\right)^2\right] dz \leq Cn^{1-2c}\int_{\mathcal{Z}\setminus B_R} \tilde{h}(\|z\| - \|f\|_{\mathcal{K}})^2\, dz$$

$$\overset{(†)}{=} Cn^{1-2c}\int_{r=R-\|f\|_{\mathcal{K}}}^{+\infty} \tilde{h}(r)^2 \mathcal{S}_{d'}(r + \|f\|_{\mathcal{K}})^{d'-1}\, dr$$

$$\leq Cn^{1-2c}2^{d'-1}\int_{r=R-\|f\|_{\mathcal{K}}}^{+\infty} \tilde{h}(r)^2 \mathcal{S}_{d'} r^{d'-1}\, dr,$$

$$\text{(for } R \geq 2\,\|f\|_{\mathcal{K}}) \tag{17}$$

where (†) switches to radial coordinates. From Lemma 5.13 of Wendland (2004) we get, for any $r > 0$:

$$|\tilde{h}(r)| \leq Cr^{s_2/2 - d'/2}\sqrt{\frac{2\pi}{r}}e^{-r}e^{|d'/2 - s_2/2|^2/(2r)}.$$

Recalling that $s_2 > d'$ by assumption we have

$$\int_{\mathcal{Z}\setminus B_R} \mathbb{E}\left[\left([\hat{\mu}^h_{f(X)} - \mu^h_{f(X)}](z)\right)^2\right] dz$$

$$\leq Cn^{1-2c}\int_{r=R-\|f\|_{\mathcal{K}}}^{+\infty} r^{s_2-2}e^{-2r}e^{(s_2-d')^2/(4r)}\, dr$$

$$\leq Cn^{1-2c}e^{\frac{(s_2-d')^2}{4(R-\|f\|_{\mathcal{K}})}}\int_{r=R-\|f\|_{\mathcal{K}}}^{+\infty} r^{s_2-2}e^{-2r}\, dr.$$

Now, $s_2/2$ being by assumption a strictly positive integer, $s_2 - 2$ is an integer. Thus, using (Gradshteyn and Ryzhik, 2007, 2.321.2)

$$\int_{r=R-\|f\|_{\mathcal{K}}}^{+\infty} r^{s_2-2}e^{-2r}\, dr = e^{-2(R-\|f\|_{\mathcal{K}})}\left(\sum_{k=0}^{s_2-2} \frac{k!\binom{s_2-2}{k}}{2^{k+1}}(R - \|f\|_{\mathcal{K}})^{s_2-2-k}\right)$$

we continue by writing

$$\int_{\mathcal{Z}\setminus B_R} \mathbb{E}\left[\left([\hat{\mu}^h_{f(X)} - \mu^h_{f(X)}](z)\right)^2\right] dz$$

$$\leq Cn^{1-2c}e^{\frac{(s_2-d')^2}{4(R-\|f\|_{\mathcal{K}})}}e^{-2(R-\|f\|_{\mathcal{K}})}(R - \|f\|_{\mathcal{K}})^{s_2-2}$$

$$\text{(for } R \geq \|f\|_{\mathcal{K}} + 1) \tag{18}$$

$$\leq Cn^{1-2c}(R - \|f\|_{\mathcal{K}})^{s_2-2}e^{-2(R-\|f\|_{\mathcal{K}})} \tag{19}$$

$$\text{(for } R \geq \|f\|_{\mathcal{K}} + \frac{(s_2 - d')^2}{4}) \, . \tag{20}$$

**Step 3: Choosing $R$ to balance the terms**

Compiling Equations (15), (16) and (19), we get:

$$\int_{\mathcal{Z}} \underset{S}{\mathbb{E}}\left[\left([\hat{\mu}^h_{f(X)} - \mu^h_{f(X)}](z)\right)^2\right] dz \leq CR^{d'}n^{-2\nu} + Cn^{1-2c}(R - \|f\|_{\mathcal{K}})^{s_2-2}e^{-2(R-\|f\|_{\mathcal{K}})}.$$

We now let $R$ depend on the sample size $n$ so that both rates be (almost) balanced. Ideally, defining $\gamma := \nu + 1/2 - c \geq 0$ and taking the log of these rates, we would thus solve

$$d' \log R = 2\gamma \log n + (s_2 - 2) \log(R - \|f\|_{\mathcal{K}}) - 2(R - \|f\|_{\mathcal{K}}) , \tag{21}$$

and stick the solution $R_s$ back into either of the two rates. Instead, we will upper bound $R_s$ and stick the upper bound into the first rate, $R^d n^{-2\nu}$, which is the one increasing with $R$. More precisely, we will now show that for large enough $n$ we can upper bound $R_s$ essentially with $2\gamma \log(n) + \|f\|_{\mathcal{K}}$, which also satisfies conditions (17), (18) and (20). This will complete the proof.

Note that $s_2 > d' \geq 1$ and $s_2 \in \mathbb{N}_+$. First assume $s_2 = 2$. Then it is easy to check that (21) has a unique solution $R_s$ satisfying $R_s \leq \gamma \log n + \|f\|_{\mathcal{K}}$ as long as $n \geq \exp\left(\frac{1-\|f\|_{\mathcal{K}}}{\gamma}\right)$.

Next, assume $s_2 > 2$. Then for $n$ large enough (21) has exactly 2 solutions and the larger of which will be denoted $R_s$. We now replace the right hand side of (21) with a lower bound $d' \log(R - \|f\|_{\mathcal{K}})$:

$$(d' - s_2 + 2) \log(R - \|f\|_{\mathcal{K}}) = 2\gamma \log n - 2(R - \|f\|_{\mathcal{K}}) , \tag{22}$$

Clearly, (22) has one (if $d' - s_2 + 2 \geq 0$) or two (if $d' - s_2 + 2 < 0$) solutions, and in both cases the larger one, $R_s^*$, satisfies $R_s^* \geq R_s$. If $d' - s_2 + 2 \geq 0$ then, for $n \geq e^{1/\gamma}$, $R_s^* \leq \|f\|_{\mathcal{K}} + \gamma \log n$, because

$$(d' - s_2 + 2) \log(\gamma \log n) \geq 0$$

Finally, if $d' - s_2 + 2 < 0$ then the smaller solution of (22) decreases to $\|f\|_{\mathcal{K}}$ and the larger one $R_s^*$ tends to infinity with growing $n$. Evaluating both sides of (22) for $R = \|f\|_{\mathcal{K}} + 2\gamma \log n$ we notice that

$$(d' - s_2 + 2) \log(2\gamma \log n) \geq -2\gamma \log n$$

for $n$ large enough, as $\log n$ increases faster than $\log \log n$. This shows that $R_s^* \leq \|f\|_{\mathcal{K}} + 2\gamma \log n$.

Thus there exists a constant $C$ independent of $n$, such that, for any $n \geq 1$:

$$\mathbb{E}\left[\left\|[\hat{\mu}_{f(X)}^{k_z} - \mu_{f(X)}^{k_z}]\right\|_{k_z}^2\right] \leq C(\log n)^{d'} n^{-2\nu} = O\left((\log n)^{d'} n^{-2\nu}\right) . \qquad \square$$