[Reviews · NeurIPS 2016]

Reviewer 1

Summary

The paper provides a consistent estimator for functions of a random variable as well as finite sample bounds, based on kernel embedding.

Qualitative Assessment

The paper is well written, original and contains some new results in the context of kernel embedding of probability measures using reproducing kernel Hilbert spaces. However, the novelty is not enough to rank the paper in the top 3% submissions.

Confidence in this Review

2-Confident (read it all; understood it all reasonably well)


Reviewer 2

Summary

Assume that we have a map f:X->Y and two kernels k_X and k_Y on X and Y, respectively. The first main result of this paper shows: if we have consistency of an empirical kernel mean embedding scheme for k_X and all distributions P on X, then this scheme is also consistent for k_Y and all image distributions P_f. In addition, some finite sample guarantees for particular kernels and maps f are provided and the case, in which X is a product space, is investigated in further detail.

Qualitative Assessment

The clarity of the presentation can certainly be improved, in particular when it comes to formally correct definitions. For example, when reading Theorem 1, it is completely unclear from its formulation, of what form the estimator actually is. One needs to refer to the proof to (only partially!!!) understand the theorem. Also, I could not really follow the proof in the case of data-dependent weights, which are presumably allowed in the Theorem?!? In addition, the footnote in this theorem is a bad idea. On a completely different page, I am really unsure whether the paper fits into NIPS. While kernel embeddings have been considered at NIPS for quite some time, there was always a clear link to machine learning in those papers. This paper lacks this connection, or at least I could not get it.

Confidence in this Review

2-Confident (read it all; understood it all reasonably well)


Reviewer 3

Summary

This paper quantifies the price of estimating the distribution of $f(X)$, when {\em dependent} Kernel mean embedding is used. As shown in Eq. (2), O(1/n) of parametric rate is achievable with KME based on i.i.d. samples. However, when "expansion-points" are dependent, the rate is not known and possibly worse. The authors quantify this gap precisely in Theorem 2. The take-home message is that when f is smooth, there is no loss, but for non-smooth f, you pay the price in sample complexity. Hence, for practitioners, this gives a guideline for deciding when to use techniques such as reduced set expansions. The paper is very well written, and I really appreciate the fact that the authors took the effort in Sections 1.1, 1.2, 1.3, and 1.4 to motivate when dependency naturally arises in KME and how memory trades off with accuracy. I learned something non-trivial that I did not know before I read this paper.

Qualitative Assessment

Quantifying the price of dependent expansion points is an interesting mathematical question. However, this paper could improve in motivating the general readers from machine learning community to get interested in a broader topic of Kernel Mean Embedding. The authors attempt to do this in the last paragraph in Section 4, relating it to Probabilistic Programming Systems, which seems to be a weak connection. AS a reader, I was curious as to where such techniques of KME and reduced set expansions can be potentially used, or are used currently in solving some application specific problems. Another disappointing aspect of the paper is that the authors did not delve deeper into the question Multiple arguments in Section 3. What is currently provided is a direct corollary of the Theorem 2, and the paper assigns too much space for something that has little information over what is already said. However, it is an interesting question to ask: given i.i.d. samples from joint distribution $(X,Y)$, what is the sample-efficient way to construct KME? (although this is outside the scope of this paper) Overall, the questions addressed in this paper is theoretically very interesting, but mainly theoretical since there is no algorithmic innovation. Further, the techniques necessary to prove the main results seem to be largely available from existing literature e.g. [Kanagawa and Fukumizu 2014] as the authors point out, which I appreciate since I could not have made the connection if the authors did not state it so clearly. Hence, this paper addresses an interesting question and gives a clean answer, but the solution ended up being simple that it lost in Novelty/originality. Typos: - On page 2, $\{(x'_j,w'_j)\}_{i=1}^n$ should be $\{(x'_j,w'_j)\}_{j=1}^N$. - On page 2, $\{(f(x'_j),w'_j)\}_{i=1}^n$ should be $\{(f(x'_j),w'_j)\}_{j=1}^N$.

Confidence in this Review

2-Confident (read it all; understood it all reasonably well)


Reviewer 4

Summary

The authors conduct asymptotic analysis of the approach by Scholkopf et al. (2015) for computing functions of random variables as represented by kernel means. Unfortunately, the proofs of the main results have some flaws.

Qualitative Assessment

Technical quality: - As mentioned above, the proofs of the main results may have some flaws. == Comments after the author feedback == Thank you for the corrections of the proofs. The new proof of Theorem 1 looks correct. I think the new proof of Theorem 2 (Solution 2) works. You may need the following slight modifications: - The parameter of the Sobolev RKHS (and the Matern kernel) should not be $b$; this constant has already been used in Line 191. Instead of $b$, let's use $s > d/2$ to denote the degree of the Sobolev RKHS, $W_2^s$. - Then the resulting rate (in the author feedback) becomes $n^{-2 (\theta b - (1-\theta) (1/2-c) )}$ with $\theta = \alpha/s$. == Comments before the author feedback == Novelty/originality: - The topic of the paper is interesting. Potential impact or usefulness: - The impact of this paper depends on that of the approach by Scholkopf et al. (2015). Therefore If this approach is proven to be useful in practice (e.g. in probabilistic programing), the aim of the current paper would be reasonable and corrected results will have significant impact. - I like Lemma 8 in Appendix and the entire proof idea of Theorem 2, as they are potentially useful for the analysis of kernel mean estimators in general. Clarity and presentation: - Overall this paper is nicely written and easy to follow. In my opinion it would be better to reproduce the results of Simon-Gabriel and Scholkopf in Appendix for the convenience of the reader. Other comments: - While the proof of Theorem 2 is not correct due to the flaw of Theorem 1 of Kanagawa and Fukumizu (2014), I believe it would be possible to replace it by some other results (e.g. Theorem 1 of http://arxiv.org/abs/1605.07254).

Confidence in this Review

3-Expert (read the paper in detail, know the area, quite certain of my opinion)


Reviewer 5

Summary

This paper presents theoretical analysis for kernel mean embedding (KME) of functions of a random variable. The analysis clarifies ... (1) for any continuous function f, if there is a consistent estimator of the mean embedding of a random valuable X, one can obtain the consistent estimator of the mean embedding of f(X) (2) convergence rate of the estimator of the mean embedding of f(X) for Gaussian kernel and sufficiently smooth f (3) in case of multiple random valuables (X,Y), the reduced set method can produce a good estimator of the mean embedding of f(X,Y)

Qualitative Assessment

This paper is clearly written and easy to follow. The analysis is sufficiently general, and its result would be applied to various application. As far as I understand, there is no major concerns about the analysis and its results shown in this paper. Some minor concerns are shown below. * In line 199, the authors state "we then almost recover the i.i.d. rate of (2)", however, I couldn't get it. Does it mean that Eq. (2) can be derived as a special case of Eq. (3) under i.i.d. assumption? Or it just means that Eq. (2) is tighter bound than Eq. (3)? * The result shown in ln. 198, which is "breaking the curse of dimensionality up to the growing logarithmic factor", is the most curious one in this paper for me. I'm interested in its experimental validation.

Confidence in this Review

2-Confident (read it all; understood it all reasonably well)


Reviewer 6

Summary

This paper proves that if a weighted sample can represent the kernel embedding of a random variable well, we can get a good estimation for the kernel embedding of a function of that random variable by passing the weighted sample to the function. The authors also provide finite sample bounds under stricter assumptions and analyze the case of functions of multiple variables.

Qualitative Assessment

The paper is well written. Contents are well motivated and the proofs in the supplementary materials are fully explained in detail. The contribution is incremental. The paper provides better estimators for estimating \mu[f(X)] and \mu[f(X,Y)] under mild assumptions, complementing the framework in Schölkopf et al's previous work. However, I am not sure what impact this work will have. There are no experiments illustrating practical applications of this work. The only experimental result of Figure 1 seems to be a replicate of the synthetic data experiment in Schölkopf et al's work. Also there are no comparisons of other estimators, although I understand that the results of Schölkopf et al's estimators may be similar to those of the estimators in this paper. Reference: B. Schölkopf, K. Muandet, K. Fukumizu, S. Harmeling, and J. Peters. Computing functions of random variables via reproducing kernel Hilbert space representations. Statistics and Computing.

Confidence in this Review

2-Confident (read it all; understood it all reasonably well)